# Identification of hypertension gene expression biomarkers based on the DeepGCFS algorithm

**Zongjin Li**[1], **Liqin Tian**[2,3], **Libing Bai**[2], **Zeyu Jia**[2], **Xiaoming Wu**[4]*, **Changxin Song**[5]*

**1** College of Science, North China University of Science and Technology, Tangshan, China, **2** School of Computer, Qinghai Normal University, Xining, Qinghai, China, **3** School of Computer, North China Institute of Science and Technology, Langfang, Hebei, China, **4** The Key Laboratory of Biomedical Information Engineering of Ministry of Education, School of Life Science and Technology, Xi'an Jiaotong University, Xi'an, Shaanxi, China, **5** Shanghai Urban Construction Vocational College, Shanghai, China

* wxm@mail.xjtu.edu.cn (XW); songcx321@163.com (CS)

**Data Availability Statement:** The data and code for this study can be downloaded at:https://github.com/lzj2051303710/Hy_DeepGCFS.

**Funding:** This research was supported by Key Laboratory of Industrial Internet of Things and Networked Control, Ministry of Education

## Abstract

Hypertension is a critical risk factor and cause of mortality in cardiovascular diseases, and it remains a global public health issue. Therefore, understanding its mechanisms is essential for treating and preventing hypertension. Gene expression data is an important source for obtaining hypertension biomarkers. However, this data has a small sample size and high feature dimensionality, posing challenges to biomarker identification. We propose a novel deep graph clustering feature selection (DeepGCFS) algorithm to identify hypertension gene biomarkers with more biological significance. This algorithm utilizes a graph network to represent the interaction information between genes, builds a GNN model, designs a loss function based on link prediction and self-supervised learning ideas for training, and allows each gene node to obtain a feature vector representing global information. The algorithm then uses hybrid clustering methods for gene module detection. Finally, it combines integrated feature selection methods to determine the gene biomarkers. The experiment revealed that all the ten identified hypertension biomarkers were significantly differentiated, and it was found that the classification performance of AUC can reach 97.50%, which is better than other literature methods. Six genes (PTGS2, TBXA2R, ZNF101, KCNJ2, MSRA, and CMTM5) have been reported to be associated with hypertension. By using GSE113439 as the validation dataset, the AUC value of classification performance was to be 95.45%, and seven of the genes (LYSMD3, TBXA2R, KLC3, GPR171, PTGS2, MSRA, and CMTM5) were to be significantly different. In addition, this algorithm's performance of gene feature vector clustering was better than other comparative methods. Therefore, the proposed algorithm has significant advantages in selecting potential hypertension biomarkers.

## 1. Introduction

Hypertension is one of the most common chronic diseases globally, severely impacting public health. Research has shown that from 1990 to 2019, the number of adults aged 30–79 with

[21567693H], the Fundamental Research Funds for the Central Universities [3142021009], the China University Industry Research Innovation Fund [2022IT230], the China University industry research innovation Fund [2021DA12008], the Technology Innovation Leading Program of Shaanxi [2023KXJ-219] to XW, Shanghai Urban Construction Vocational College School-level Project [cjky202203] to SC, and the Qinhai IoT Key Laboratory [2017-ZJ-Y21] to LT. The funders played a critical role in study design, data collection and analysis, decision to publish, and preparation of the manuscript.

**Competing interests:** The authors have declared that no competing interests exist.

hypertension increased from 650 million to 1.28 billion. Almost half of these individuals are unaware of their condition [1]. Hypertension is considered a significant risk factor for cardiovascular diseases, kidney diseases, and other associated complications. Clinical diagnosis and assessment of hypertension primarily rely on measuring blood pressure values and some biochemical indicators (urea nitrogen, cholesterol, triglycerides, etc.). However, many factors can influence blood pressure values, exhibit diurnal variability, and be subject to measurement errors. Therefore, it is necessary to find reliable biomarkers to assist in classifying, identifying, or treating hypertension.

Aberrant regulation of gene expression plays a crucial role in the pathogenesis of hypertension. Gene expression data obtained through high-throughput sequencing techniques have emerged as the primary data type for exploring the mechanisms of hypertension, providing a foundational basis for identifying biomarkers. Early methods for biomarker identification relied on Differential Expression Gene (DEG) analysis, employing statistical approaches. Commonly used tools include DESeq, Limma, and edgeR, which are R packages [2]. This method is still widely used in current research. For instance, Gao et al. [3] when exploring potential biomarker genes associated with hypertension, firstly, carried out DEG through the limma package, then obtained the interactions network between genes through the STRING platform, subsequently, applied three centrality algorithms (degree centrality, median centrality, and subgraph centrality) in the CytoNCA plugin to identify the hub genes in the network, and finally revealed the critical roles of Cyp4b1, Cyp4a31, Loxl2, and TFs, Esr1, Pparg, and Rxrg in the development of hypertension. However, DEG, by setting specific thresholds, may overlook hub genes. Indeed, many genes involved in the pathologic process of hypertension may be minimally expressed but critically important, and these genes are likely to be excluded under strict threshold criteria. Furthermore, this method often overlooks structural and attribute information among genes, which are crucial for unraveling the pathogenesis of hypertension and identifying functionally critical genes.

Researchers increasingly recognize that molecules within biological cells do not exist in isolation but form intricate networks. Therefore, exploring disease biomarkers based on co-expression modules has become a research hotspot. A popular method for this purpose is Weighted Gene Co-expression Network Analysis (WGCNA), which introduces the concept of soft threshold and realizes the construction of the network based on the expression similarity between genes [4]. It offers unique advantages in dealing with multi-sample biomarker screening. Li et al. [5] utilized WGCNA to construct co-expression networks of mRNA and miRNA in hypertension, uncovering two key gene modules and one key miRNA module. By calculating indicators such as module significance, module membership, and network centrality, they identified 12 gene markers and ten miRNA markers for hypertension. However, this method overlooks potential local expression relationships that may exist only among specific genes and inadequately handles higher-order associations, potentially missing some deeper connections within complex biological systems.

With the rapid development of artificial intelligence technologies, feature selection methods, owing to their ability to significantly reduce the dimensionality of feature space while preserving the accurate representation of selected features from the original data, have been widely employed in identifying genetic biomarkers for hypertension [6,7]. For example, Reel et al.[8] in exploring biomarkers distinguishing different types of hypertension, they employed random forest and correlation-based feature selection methods to filter multidimensional omics data features, and they utilized machine learning classifiers to study the classification of five different types of hypertension diseases. Ultimately, they identified has-miR-15a-5p and two plasma metabolites (C9 and PC ae C38:1) as biomarkers. Although this study exhibits advantages in integrating multi-omics data through machine learning methods, it solely relies

on machine learning methods for biomarker identification. It fails to consider the topological structure among features fully. Jiang et al.[9] in their study of genetic diagnostic biomarkers for pulmonary arterial hypertension, they first screened out the differential genes by DEG, then used LASSO regression and SVM, two feature selection methods, to identify biomarkers. Eventually, they selected CALD1 and SLC7A11, jointly identified by the two methods, as biomarkers. Although the feature selection methods perform well in identifying hypertension biomarkers, there are some limitations. Specifically, filtered feature selection is computationally fast and easy to understand but may ignore complex interactions between features. Wrapper feature selection, although capable of capturing indirect relationships between features, may suffer from overfitting due to reliance on specific models and training processes and typically entails significant computational overhead. Embedded feature selection, while adapting well to model structures and considering interactions among features, may increase the risk of overfitting, sacrificing interpretability, and escalating computational complexity due to its high dependency on specific models. Thus, it presents challenges in balancing feature relationship comprehension and generalization ability. Although hybrid feature selection methods may mitigate the drawbacks of various feature selection methods to some extent, most existing methods, when handling hypertension gene expression data, assume independence among genes, neglecting the intricate regulatory network relationships among genes, which are crucial for effectively identifying biomarkers. Therefore, utilizing feature selection methods for hypertension genetic biomarker identification still entails certain limitations.

With the advancement of graph representation learning, many researchers have applied it to the analysis of gene expression data. Graph representation learning aims to map nodes in a graph to a low-dimensional vector space while preserving the original information, thereby facilitating research on identifying gene biomarkers. Traditional methods of graph representation learning involve dimensionality reduction techniques. Currently, researchers have proposed many methods of graph representation learning, such as DeepWalk[10] and Node2vec [11], which capture similarities between nodes based on random walks and word embedding techniques, and SDNE[12], which utilizes neural networks to learn low-dimensional representations of nodes while preserving the network's topological structure. Although these methods demonstrate powerful capabilities in capturing local neighborhood structures of nodes and learning implicit relationships between nodes and can effectively improve the performance and efficiency of subsequent biomarker identification tasks, they still lack complete modeling of the global topological structure of gene graph networks and ignore node attribute information and edge weight information, thereby limiting their potential in comprehensively understanding and mining the deep structure-function relationships of gene expression networks. To address the issues above, Graph Neural Networks (GNN), a new method of graph representation learning, has emerged to provide a new perspective and tool for studying graph data [13]. GNN can effectively capture complex relationships in the network and demonstrate powerful representation learning and inference capabilities[14,15]. These have led to the widespread use of GNN in various research fields, such as social sciences, protein-protein interaction networks, knowledge graphs, and biomarker identification[16–18]. However, there is almost no research based on GNN in the specific task of identifying hypertension biomarkers.

To better identify biomarkers in gene expression data of hypertension, we propose a novel algorithm named Deep Graph Clustering Feature Selection (DeepGCFS), as illustrated in Fig 1. The algorithm uses the graph network structure to represent gene relationships, combining GNN, clustering, and feature selection methods. Specifically, the algorithm utilizes interaction relationships between genes obtained from the STRING database as prior knowledge and calculates gene similarities to co-construct a more stable gene graph network. Then, a new

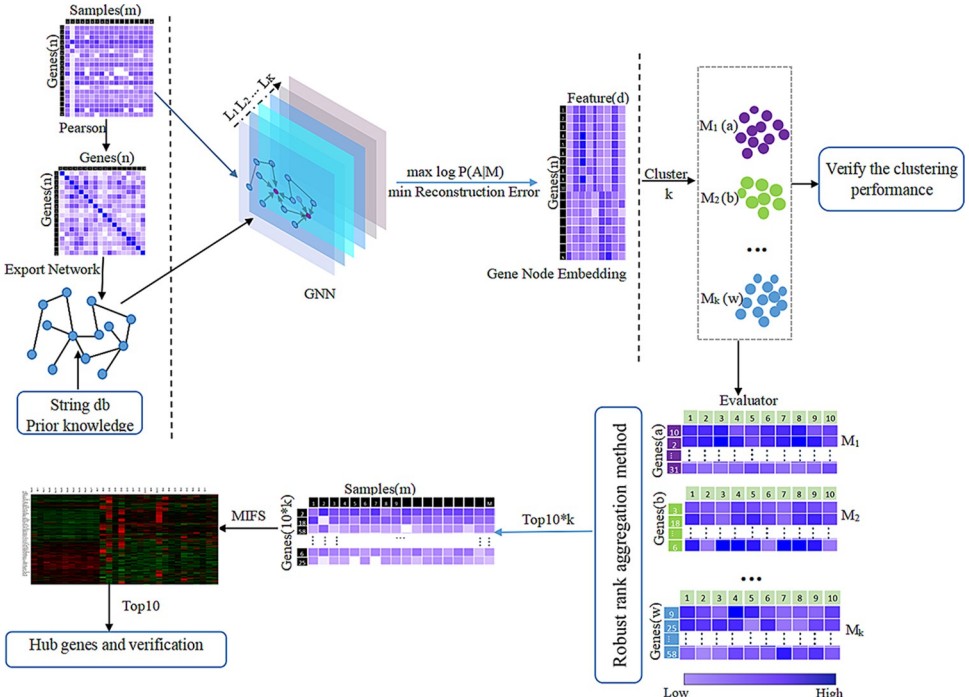

**Fig 1. Algorithm framework.** n represents the number of genes, m represents the number of samples, d represents the dimensionality of features obtained through GNN representation learning, k represents the number of gene clusters modules, M represents gene modules, M1(a) denotes that the number of genes in module1 is a, Evaluator denotes then feature selection methods for gene evaluation, and MIFS represents Mutual Information Feature Selection.

objective function is formulated, and gene representation learning is conducted based on GNN methods. The Hybrid Clustering Method (HCM) method is proposed for clustering. Finally, the algorithm integrates ten feature selection methods to obtain more accurate biomarkers. The effectiveness of the DeepGCFS algorithm and the accuracy of the identified gene biomarkers are validated through the significance of difference, classification performance, and external dataset verification.

# 2. Materials and methods

## 2.1 Data source and data preprocessing

The hypertension RNA sequencing (RNA-seq) data SRP447196 was obtained from the SRA database (https://www.ncbi.nlm.nih.gov/sra/) containing 20 hypertension patients and ten healthy individuals. We conducted quality control of RNA-seq data using software such as bbmap, fastx_trimmer, and fastqc in an Ubuntu (22.04.3) environment. Sequence alignment was carried out using the Hisat2 software, with hg38 as the reference genome and the Homo_sapiens.GRCh38.100.gtf.gz (http://ftp.ensembl.org/pub/current_gtf/homo_sapiens/) file was used as the annotation file. The gene expression matrix was calculated using the featureCounts program, and then preprocessing was continued, such as deduplication, handling missing values, and normalization in the R programming language (V4.3.1). The validation dataset in this study was GSE113439 from the GEO database (http://www.ncbi.nlm.nih.gov/geo/), which consists of 11 normal samples and 15 individual samples of pulmonary hypertension.

## 2.2 Establishment of genes graph structure

First, we chose the Pearson correlation coefficient to calculate the similarity between each pair of genes, as shown in Eq (1). Since the network constructed by the similarity between genes is a complete graph, the high-density connections inherent in complete graphs may lead to significant computational overhead, increased noise, and redundant information, consequently affecting the effective learning and generalization performance of GNN. Moreover, the complete graph does not reflect the reality of gene expression. Therefore, it is necessary to prune the complete graph by filtering edges or nodes. In this study, we performed pruning on edges between genes. Specifically, we removed edges with weights below a certain threshold. For the selection of this threshold, we set it to a value obtained in the absence of isolated points. The resulting pruned graph network is denoted as $e^{G_p}$. Meanwhile, we utilized the physical interaction information from the STRING database as prior knowledge of gene interactions. Specifically, genes were imported into the STRING plugin database embedded within the Cytoscape software to obtain a network of gene interactions (with interaction scores greater than 0.4), denoted as $e^{G_s}$. Then, the $e^{G_p}$ and $e^{G_s}$ were merged, and duplicate edges were removed to form the final graph network, denoted as G. The specific process is detailed in Algorithm 1. The weights of edges between genes were computed according to Eq (2).

$$r\left(v_i, v_j\right) = \frac{\text{cov}(v_i, v_j)}{\sigma v_i \sigma v_j} = \frac{\sum_{\mu=1}^{m}(v_{i\mu} - \bar{v}_i)(v_{j\mu} - \bar{v}_j)}{\sqrt{\sum_{\mu=1}^{m}(v_{i\mu} - \bar{v}_i)^2}\sqrt{\sum_{\mu=1}^{m}(v_{j\mu} - \bar{v}_j)^2}} \tag{1}$$

Where $\bar{v}_i$ and $\bar{v}_j$ represent the average expression levels of genes $v_i$ and $v_j$ cross m experimental conditions, respectively. $v_{i\mu}$ and $v_{j\mu}$ represent the expression levels of $v_i$ and $v_j$ in the $\mu$-th experimental condition, respectively.

$$W = \begin{cases} \text{mean}(W(e^{G_p}_{(v_i,v_j)}) + W(e^{G_s}_{(v_i,v_j)})), & (v_i, v_j) \in (G_p \cap G_s) \\ \dfrac{W(e^{G_p}_{(v_i,v_j)})}{2}, & (v_i, v_j) \in G_p \cap (v_i, v_j) \notin G_s \\ \dfrac{W(e^{G_s}_{(v_i,v_j)})}{2}, & (v_i, v_j) \in G_s \cap (v_i, v_j) \notin G_p \end{cases} \tag{2}$$

In graph $G = \{V,E,W\}$, $V$ represents the set of all nodes, $E$ represents the set of all edges, and $W$ is the set of weights for all nodes. In the gene expression matrix, each column represents a gene, each row represents a sample, and the matrix values denote the expression levels of genes in the samples. The sample set is defined as $S = \{s_1, s_2, \ldots s_m\}$, and the gene set is $V = \{v_1, v_2, \ldots, v_n\}$. Then each sample can be represented as $S_i = \{v_1^i, v_2^i, \ldots v_n^i\}$. Since each gene is treated as a node $v$ in the graph, the initial feature vector for each node is composed of the feature values of each gene over the samples, denoted by $h_{v_i}^0 = [s_1^{v_i}, s_2^{v_i}, \ldots s_m^{v_i}]$.

```
Algorithm 1. Construction of Network G.
Input: df_1, df_2; //Data frames that represent the graphs e^Gp and e^Gs,
respectively, composed of columns 'fromNode', 'toNode', and 'weight'.
gene_names. //list of genes under investigation.
Output: G.
1: df2_filtered = []
2: for each row in df_2:
3:   if row['fromNode'] in gene_names and row['toNode'] in gene_names
then:
4:     add row to df2_filtered;
5:   end if
```

```
6: end for
7: merged_df ← df_1.copy() // copy the entire contents of df_1 into
merged_df
8: for each row2 in df2_filtered:
9:   is_common_edge ← False
10:   for each row1 in merged_df:
11:     if row1['fromNode'] = = row2['fromNode'] and row1['toNode'] =
= row2['toNode'] then:
12:        row1['weight'] + = row2['weight'] // If edges exist in mer-
ged_df, the weights are updated
13:        is_common_edge←True
14:        break
15:     end if
16:   end for
17:   if not is_common_edge then:
18:     add row2 to merged_df //If the edge does not exist in mer-
ged_df, it is added.
19:   end if
20:end for
21:merged_df['weight'] / = 2.0;
22: G←merged_df;
23: return G.
```

## 2.3 Information propagation and weighted aggregation

Before information propagation, it is necessary to sample the nodes. We take node pairs with edge connections as positive samples and those without edge connections as negative samples. We select a certain percentage of positive and negative edges in each training batch to balance the positive and negative samples. To ensure that different types of neighbors aggregate the same node in different batches, we sample the neighbors of each gene node to be calculated.

Information propagation is a crucial component of GNN. Traditional graph neural networks assume equal contributions of each edge to the aggregation process. However, the biological network constructed using gene expression data belongs to the weighted network, where two nodes connected by higher-weighted edges should have a higher mutual influence [17]. Therefore, we introduce weighted aggregation functions. Before aggregation, the weights of all sampling edges are normalized using Eq (3) to ensure that the sum of weights of all sampling edges is 1. During the aggregation of the vectors of neighboring nodes, each neighbor node vector is multiplied by the normalized weight of its corresponding edge, as computed by Eq (4).

$$weight_{norm}^{(v_i, v_j)} = \frac{weight_{(v_i, v_j)}}{\sum_{v_l \in V} weight_{(v_i, v_l)}} \tag{3}$$

$$h_V^K = \text{AGGREGATE}_K(\{h_{v_i}^{K-1} \cdot weight_{norm}^{(v_i, v_j)}, \forall v_i \in V\}) \tag{4}$$

Where $V$ represents the set of all neighboring nodes of node $v_i$, $h_V^K$ represents the $K$-th layer hidden state vector of node $V$, weightnorm represents the normalized weight of each edge, and $\text{AGGREGATE}_K(\cdot)$ represents the feature aggregation function. The model uses an average aggregation strategy, summing the features of each node and those of its neighbors, and averaging the features.

Next, the state vector of each node's current layer is concatenated with the state vector of the previous layer, and the trained weight matrix $W^K$ is used for linear transformation. Finally,

a nonlinear transformation is performed through σ(·), as shown in Eq (5).

$$h_{v_i}^K = \sigma(W^K \cdot \text{CONCAT}(h_{v_i}^{K-1}, h_V^K)) \tag{5}$$

Where σ(·) is the Relu activation function, $W^K$ is the weight matrix of the $K$-th layer, CON-CAT(·) r represents the concatenation function, and $h_{v_i}^K$ is the vector of node $v_i$ in $K$-th layer. To prevent overfitting and speed up training, Dropout and Batch Normalization mechanisms are used in each layer of the neural network.

## 2.4 Link function

Since the genes in a dataset of hypertension do not contain labels, the supervised methods cannot be applied to node representation learning. Therefore, we introduce link prediction and self-supervised learning ideas to construct a new loss function.

Studies have shown that if two nodes belong to the same community or have the same label, they are more likely to share an edge in the graph[19]. In other words, nodes within a community or with the same label are more likely to be connected. Assuming M is a given community affiliation matrix, we calculate the probability that any two nodes ($v_i$,$v_j$) belong to the same community in the graph by Eq (6). The community affiliation matrix M is a latent variable generating the current network. The true M should maximize the p(Ω|$M$) [20]. Therefore, the first part of the loss function is depicted in Eq (6).

$$\phi_{v_i v_j} = M_{v_i} M_{v_j}^T \tag{6}$$

Then, we utilized Eq (7) as the connectivity function to quantify the probability of the edges existing between two nodes. In this framework, node pairs with edges are considered positive examples, while those without edges are considered negative. Given that edges in real graph, models are often sparse, with the number of negative samples far exceeding positive ones, we selectively sample positive and negative edges in each training batch to balance their influence. To address this issue further, we employ the binary cross-entropy loss function for optimization. Additionally, to prevent information leakage during training, those positive edges used to compute the loss are temporarily removed from the current batch of the original graph in each batch processing, ensuring their exclusion from sampling and message aggregation, thereby ensuring the accuracy and effectiveness of training. The first part of the loss function in this study is illustrated in Eq (8).

$$\sigma(\phi) = \frac{1}{1 + \exp(-\phi_{v_i v_j})} \tag{7}$$

$$L_l = -E_{(v_i, v_j) \sim P_E} \log(p(\Omega_{v_i, v_j} = 1 | M_{v_i}, M_{v_j})) - E_{(v_i, v_j) \sim P_N} \log(p(\Omega_{v_i, v_j} = 1 | M_{v_i}, M_{v_j})) \tag{8}$$

Where $P_E$ and $P_N$ represent the uniform sampling of positive and negative samples, respectively. $E_{(v_i, v_j) \sim P_E}$ and $E_{(v_i, v_j) \sim P_N}$ represent the influence weights of the positive and negative samples on the loss function. $M_{v_i}$ and $M_{v_j}$ represent the communities to which node $v_i$ and $v_j$ belong.

Self-supervised learning is a machine learning method that learns useful representations from unlabeled data and enhances the model's generalization capability and robustness. To comprehensively capture interaction patterns among genes, we introduced the concept of self-supervised learning[21]. The second part of the loss function is constructed to improve model performance and provide deeper biological insights by minimizing the cross-entropy between

the reconstructed adjacency matrix and the true adjacency matrix. The second part of the loss function is as follows:

$$L_{re} = \sum_{v_i \in V, v_j \in V} (-A_{v_i,v_j} \log(\hat{A}_{v_i,v_j}) - (1 - A_{v_i,v_j}) \log(1 - \hat{A}_{v_i,v_j})) \tag{9}$$

$$\hat{A}_{v_i,v_j} = \sigma(z_{v_i}^T z_{v_i}), z_{v_i} = Z v_i \tag{10}$$

Where $A$ represents the adjacency matrix, $Z$ corresponds to the feature vectors obtained through representation learning for gene nodes, the $i$-th row of $Z$ represents the representation $z_i$ of node $v_i$. $\hat{A}_{v_i,v_j}$ is the predicted probability of the link $(v_i, v_j)$, and $\sigma(\cdot)$ is the sigmoid function.

Therefore, the final loss function is as shown in Eq (11).

$$L = L_l + L_{re} \tag{11}$$

## 2.5 Module detection and compared with other clustering methods

After obtaining the feature vectors of gene nodes, we used clustering algorithms for module division. To mitigate subjectivity in the selection of the module number ($k$), we proposed an HCM method. The method first employs the Dynamic Hybrid Tree Cut (DHT) to identify the most suitable number of modules and then performs clustering using k-means. DHT considers the changes of genes under different conditions and adaptively adjusts connectivity during clustering, automatically determining the optimal number of modules[22,23]. K-means algorithm is simple, efficient, and easy to interpret. We adopt this clustering strategy for module partitioning. The HCM method is described in Algorithm 2.

```
Algorithm 2. HCM Algorithm.
Input: M_d //Gene expression matrix.
Output: K_c // The results of clustering.
1: sim←cor(M_d); //The correlation matrix sim between genes was calcu-
lated by Pearson correlation coefficient.
2: htree←hclust(as.dist(1-sim), meth = "average"); //Obtain the tree
graph from the hierarchical clustering using the hierarchical cluster-
ing hclust method.
3:set minModuleSize = 30; //Minimum number of genes contained in a
module.
4: k←num(cutreeDynamic(dendro←htree, distM←sim, deepSplit←2,
pamRes-
pectsDendro←FALSE, minClusterSize = minModuleSize)); // Determine the
optimal number(k) of modules.
5: centroids ←sample_n(Data, k); //Randomly initialize k cluster
centroids.
6: while (true):
7: distances←apply(Data, 1, function(x) euclidean_distance(x,
centroids));
//Calculate the distance of each gene to the center of mass.
8: K_c ← apply(distances, 1, which.min); //Assign each gene to the
cluster with the nearest centroid.
9:  if K_c = = previous_clusters then: //Check whether met the stop
conditions.
10:  break
11: end if
12: for i in 1:k:
```

```
13:  centroids[i,]← mean(Data[clusters = = i,], na.rm = TRUE);
14:  previous_clusters←clusters;
15: end for
16: return clusters
17:end while
```

To validate the proposed method's clustering performance, we compared the clustering results with seven other clustering algorithms, including k-means, Spectral cluster, DHT, FCM, deepwalk, node2vec, and SDNE. Since genes are unlabeled, internal evaluation metrics like the Silhouette Index (SI), Calinski-Harabasz Index (CHI), and Davies-Bouldin Index (DBI) are used. The SI ranges from -1 to 1, with higher values indicating better clustering; the higher the CHI value, the better the clustering, and the lower the DBI value, the better the clustering[24,25].

## 2.6 Enrichment analysis and visualization of gene modules

We conducted Gene Ontology analysis (GO) and Kyoto Encyclopedia of Genes and Genomes analysis (KEGG) on the gene modules to understand the functional and pathway information. We used a hypergeometric test, employing the clusterProfiler package and org.Hs.eg.db database, to test the significance of functions in a group of genes, with P<0.05 considered biologically meaningful.

Then, to explore the regulatory status of genes within the modules and their interactions, we initially inputted gene modules into STRING to obtain information on gene connections. Meanwhile, DEG analysis is to acquire regulatory information between genes. Then, we visualized the connectivity of genes within modules and their regulatory information using Cytoscape.

## 2.7 Identification for gene biomarkers

Each gene module contains multiple genes. Since these modules are obtained through clustering, we can consider genes in each module to be highly redundant. We use ten different feature selection methods to evaluate and rank genes in each module to select the most representative feature gene subset for each module. These include MIFS, Maximal Information Coefficient (MIC), Analysis of Variance (ANOVA), feature selection based on feature weights (ReliefF), Gradient Boosting Decision Tree (GBDT), Random Forest (RF), Lasso regression, Ridge regression, Linear regression (LR), and Decision Tree (DT). In RF, we set the maximum tree depth to 3 and the number of trees to 100; in Lasso regression, we set the alpha parameter, representing the strength of the regularization term, to 0.01; ridge regression. For other feature evaluation methods, we use the default parameters provided by the sklearn library.

Each feature evaluation method assesses features in each module, generating ten feature ranking lists for each module: $F_{MK} = [f_{Mk1}, f_{Mk2}, \ldots, f_{Mk10}]$. Then, we use the Robust Rank Aggregation (RRA) method to fusion the ten rankings for each module, outputting the top ten features after fusion. RRA is a widely used feature fusion method that synthesizes multiple evaluation results to produce the optimal feature ranking[26]. After performing this operation on each module, we obtain a subset containing $10*k$ features. Finally, considering that the MIFS method can effectively reveal the complex nonlinear relationship between gene expression and phenotypic changes, it does not require predefined parameters, thus reducing the potential impact of human factors on the results. Therefore, we selected MIFS to further optimize the genes involved in the feature subset. We selected the top 10 genes as potential biomarkers based on this refinement.

## 2.8 Validating the differential significance of biomarkers

To validate the biological significance of these ten biomarkers, we first conducted a t-test to examine whether they exhibit significant differences between the disease and control groups ($p < 0.05$). Then, we visualized the biomarkers through clustering heatmap analysis.

## 2.9 Classification performance analysis of biomarkers

We use DT as the classifier to validate the gene biomarkers' classification performance. The classification performance of the gene biomarkers is validated by the leave-one-out method. We use the Receiver Operating Characteristic (ROC) curve and Area Under the Curve (AUC) as evaluation metrics. AUC is commonly used to evaluate the performance of a binary classifier, and a value closer to 1 indicates better classification performance. The ROC curve is a plot with the True Positive Rate (TPR) on the y-axis and the False Positive Rate (FPR) on the x-axis. The formulas for calculating TPR and FPR are as follows[27]:

$$TPR = \frac{TP}{TP + FN} \tag{12}$$

$$FPR = \frac{FP}{FP + TN} \tag{13}$$

Where *TP* and *TN* are the numbers of correctly identified positive and negative samples, respectively, *FN* and *FP* are the numbers of incorrectly identified positive and negative samples.

## 2.10 External database set validation

We performed the external validation of the gene biomarkers using the GSE113439 dataset. We evaluated the classification performance of the biomarkers on this dataset. We assessed the classification performance of each gene individually in the dataset. In addition, we validate the significance of the differences in each gene using the Wilcoxon rank-sum test.

# 3. Results

## 3.1 Data preprocessing

After preprocessing the hypertension RNA-seq data, we obtained 22,462 genes. Considering the difficulty distinguishing low-expression genes from noise from a biological perspective [28,29], we filtered the genes based on the dataset's coefficient of variation (CV) and average values. Genes with an average intensity and a CV greater than 0.2 were selected for subsequent analysis, yielding 4,448 genes.

## 3.2 Graph structure establishment

We first calculated the correlation between genes according to Eq (1) and deleted the edges with a correlation of less than 0.556526 while ensuring no outliers. Thus, a gene network with 2068038 edges was preliminarily constructed. Then, 4448 gene nodes were imported into the STRING database, and a network with 21191 edges was obtained. Finally, according to the merging strategy described in Algorithm 1, a final network with 2076049 edges was obtained.

## 3.3 Gene module analysis

We used torch and DGL frameworks to construct GNN models, use gene expression data as feature attributes of nodes, and input their graph network structure into GNN for node

representation learning. We found a three-layer GNN can obtain better feature vectors. Each neural network layer used Dropout (set at 0.5) and Batch Normalization mechanisms. The model optimization was carried out using the Adam optimizer.

We partitioned gene modules using the HCM method. First, we employed the HDT method to automatically determine the optimal number of modules, denoted as k, for the gene node representation matrix, resulting in k = 22. Then, K-means clustering analysis was conducted to assign genes to the 22 modules.

In clustering comparison, all methods used the Pearson correlation coefficients for similarity measurement for fairness. As the initial module number k is a crucial parameter for the algorithm, it was first obtained automatically using DHT and set as the module number for the other seven clustering algorithms. DHT divided genes into 22 classes, so we set the module number to 22. Table 1 shows the clustering evaluation results, with all bold values representing the best method. The results show that the proposed method outperforms all three evaluation metrics.

## 3.4 Enrichment analysis

We conducted functional enrichment analysis on gene modules and found five particularly prominent gene modules that play a role in different aspects of hypertension. As shown in Table 2, the top three most significant GO terms in these five modules are listed (p<0.05, with biological significance). In addition, to more intuitively show the GO analysis results of the modules, we conducted visualization processing on these five modules. Among them, Fig 2 shows the visualization of GO enrichment results of module 1, and the GO analysis visualizations of the other four modules are shown in S1A–S1D Fig.

We also visualized the enrichment pathways of module 1, as shown in Fig 3. Module 1 is significantly involved in a series of pathways that may affect the occurrence of hypertension, such as the PPAR signaling pathway, aldosterone-regulated sodium reabsorption, cholesterol metabolism, insulin secretion, and cardiac muscle contraction. The KEGG visualization results of the other four modules are shown in S2A–S2D Fig.

## 3.5 Visualization of gene modules

We respectively input the five gene modules of interest in the last section into STRING to obtain the connection information between genes. Then, the obtained node connection information and the corresponding gene regulation information were input into Cytoscape for visualization. Fig 4 shows the visualization result of module 1, and the visualization results of other modules are shown in S3A–S3D Fig. Each node in the figure represents a gene, and each

**Table 1. Clustering performance comparison.**

|                  | SI(%)  | CHI       | DBI  |
|------------------|--------|-----------|------|
| K-means          | 20.41  | 961.64    | 1.62 |
| Spectral_cluster | 16.42  | 786.51    | 1.71 |
| DHT              | 14.58  | 521.46    | 2.17 |
| FCM              | 1.38   | 678.01    | 3.37 |
| Deepwalk         | 24.10  | 522.83    | 1.25 |
| Node2vec         | 25.01  | 654.70    | 1.22 |
| SDNE             | 83.03  | 10123.78  | 0.96 |
| Our method       | **83.57** | **18348.61** | **0.87** |

**Table 2. The most significant GO terms and p-values for each of the five modules.**

|  | ONTOLOGY | Description | pvalue |
|---|---|---|---|
| Module1 | BP~GO:0008217 | regulation of blood pressure | 1.10E-06 |
|  | BP~GO:0006644 | phospholipid metabolic process | 9.98E-05 |
|  | BP~GO:0019915 | lipid storage | 1.13E-04 |
|  | CC~GO:0034703 | cation channel complex | 2.22E-04 |
|  | CC~GO:1902495 | transmembrane transporter complex | 8.79E-04 |
|  | CC~GO:1990351 | transporter complex | 1.22E-03 |
|  | MF~GO:0046873 | metal ion transmembrane transporter activity | 3.03E-04 |
|  | MF~GO:0048018 | receptor ligand activity | 6.71E-04 |
|  | MF~GO:0051428 | peptide hormone receptor binding | 1.46E-03 |
| Module2 | BP~GO:0002002 | regulation of angiotensin levels in blood | 9.05E-05 |
|  | BP~GO:0042447 | hormone catabolic process | 1.33E-04 |
|  | BP~GO:0045216 | cell-cell junction organization | 2.42E-04 |
|  | CC~GO:0097136 | Bcl-2 family protein complex | 1.63E-02 |
|  | CC~GO:0031838 | haptoglobin-hemoglobin complex | 1.79E-02 |
|  | CC~GO:0099092 | postsynaptic density, intracellular component | 2.11E-02 |
|  | MF~GO:0008236 | serine-type peptidase activity | 2.46E-04 |
|  | MF~GO:0017171 | serine hydrolase activity | 2.67E-04 |
|  | MF~GO:0004252 | serine-type endopeptidase activity | 2.63E-03 |
| Module3 | BP~GO:0045637 | regulation of myeloid cell differentiation | 2.08E-04 |
|  | BP~GO:0051917 | regulation of fibrinolysis | 2.83E-04 |
|  | BP~GO:0045638 | negative regulation of myeloid cell differentiation | 3.07E-04 |
|  | CC~GO:0032797 | SMN complex | 1.32E-02 |
|  | CC~GO:0000786 | nucleosome | 1.35E-02 |
|  | CC~GO:0031314 | extrinsic component of mitochondrial inner membrane | 1.59E-02 |
|  | MF~GO:0004252 | serine-type endopeptidase activity | 1.54E-03 |
|  | MF~GO:0008236 | serine-type peptidase activity | 2.12E-03 |
|  | MF~GO:0017171 | serine hydrolase activity | 2.25E-03 |
| Module4 | BP~GO:1905563 | negative regulation of vascular endothelial cell proliferation | 1.01E-04 |
|  | BP~GO:0001937 | negative regulation of endothelial cell proliferation | 1.80E-04 |
|  | BP~GO:0016042 | lipid catabolic process | 1.98E-04 |
|  | CC~GO:0098992 | neuronal dense core vesicle | 4.39E-03 |
|  | CC~GO:0031045 | dense core granule | 9.24E-03 |
|  | CC~GO:0005891 | voltage-gated calcium channel complex | 1.62E-02 |
|  | MF~GO:0001664 | G protein-coupled receptor binding | 1.01E-03 |
|  | MF~GO:0008528 | G protein-coupled peptide receptor activity | 2.47E-03 |
|  | MF~GO:0036041 | long-chain fatty acid binding | 2.55E-03 |
| Module5 | BP~GO:1902287 | semaphorin-plexin signaling pathway involved in axon guidance | 7.86E-05 |
|  | BP~GO:1902285 | semaphorin-plexin signaling pathway involved in neuron projection guidance | 1.39E-04 |
|  | BP~GO:0048535 | lymph node development | 2.27E-04 |
|  | CC~GO:0002116 | semaphorin receptor complex | 7.86E-04 |
|  | CC~GO:0005759 | mitochondrial matrix | 9.27E-04 |
|  | CC~GO:0042641 | actomyosin | 2.13E-03 |
|  | MF~GO:0004602 | glutathione peroxidase activity | 5.99E-04 |
|  | MF~GO:0017154 | semaphorin receptor activity | 1.15E-03 |
|  | MF~GO:0000287 | magnesium ion binding | 2.59E-03 |

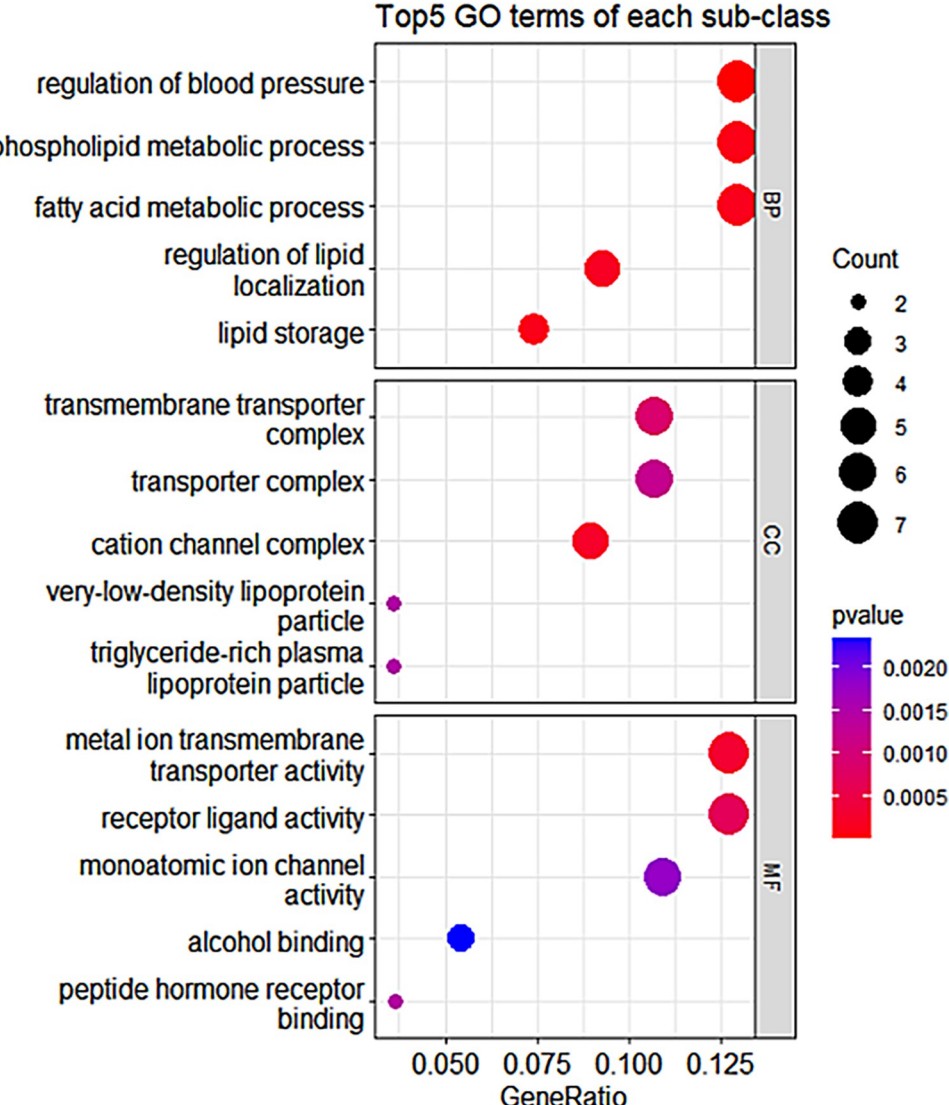

**Fig 2. GO analysis bubble plot for module 1.**

edge represents the interaction relationship between two genes. Up-regulated genes in the module are represented in red and down-regulated genes in green.

## 3.6 Biomarkers analysis

We identified the biomarkers of hypertension through the feature selection method. First, genes in each module were ranked using ten feature selection methods. Then, the ten ranking results in each module were fused using the RAA method. The top 10 genes from each module were selected as critical, totaling 220 genes. Finally, these 220 genes were re-ranked using the MIFS method, and ten gene biomarkers were selected, as shown in Table 3.

To demonstrate significant differences in gene biomarkers between disease and control groups, we performed t-test analysis on these biomarkers. The results in Table 4 show significant differences in those biomarkers. This confirms the efficacy of our methods.

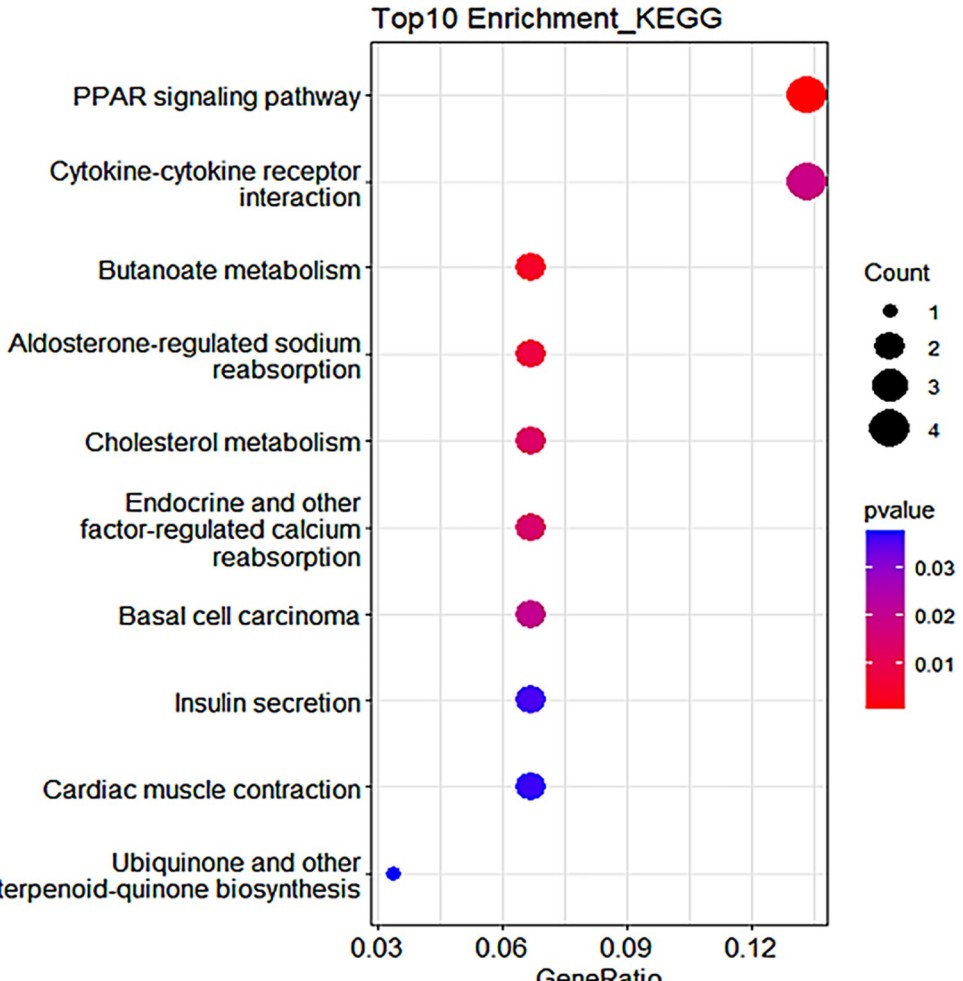

**Fig 3. Pathway analysis bubble plot for module 1.**

Fig 5 shows the heatmap of gene biomarkers across different groups. It can be observed these biomarkers exhibit high discriminative power across different sample groups, significantly distinguishing samples from different groups. This indicates the significance of these biomarkers.

## 3.7 Classification performance of biomarkers

When using all biomarkers as features, we observed an AUC value of 97.50%, indicating excellent classification performance of these biomarkers, as shown in Fig 6. When individually classifying each gene as a feature, we found that the AUC value for each gene's classification performance was above 92%, as detailed in S4A–S4J Fig.

## 3.8 Algorithm performance comparison

To validate the proposed algorithm's performance, we compared it with existing methods for hypertension biomarker identification, using the AUC as the evaluation metric. Specific results are presented in Table 5 and Fig 7. All the models listed in Table 5 have been benchmarked on

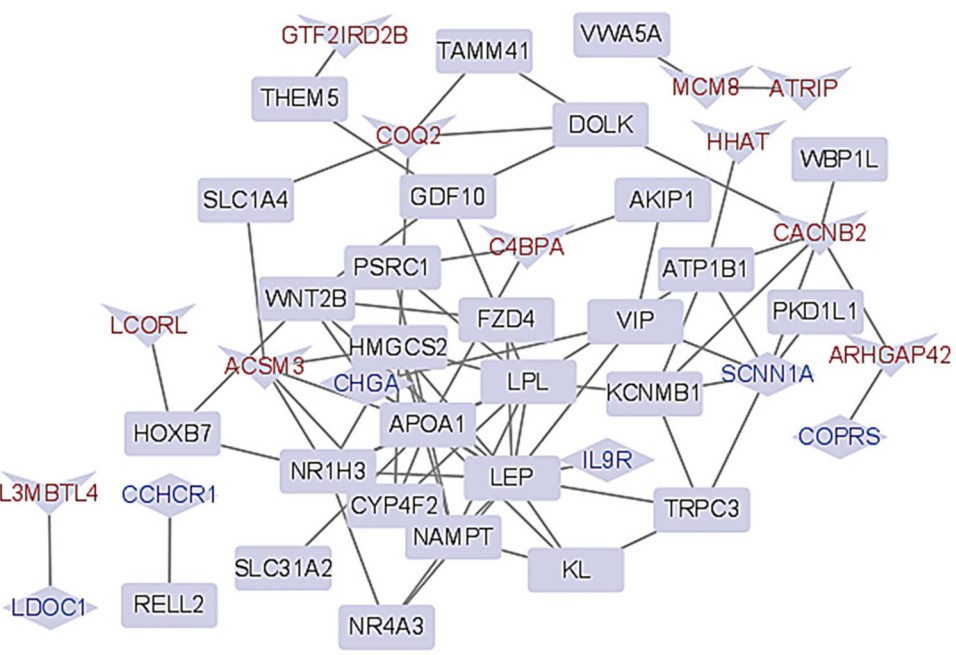

**Fig 4. Visualization for the module 1.**

the same dataset, ensuring consistency and fairness in the comparison. The experimental findings indicate that the proposed method outperforms other literature.

## 3.9 External database set validation

To further validate the effectiveness of the gene biomarkers, we used the GSE113439 dataset as an external dataset for validation. When all biomarkers were used as features, the classification performance remained satisfactory, with an AUC of 95.45%, as shown in Fig 8. When individual genes were used as features, their AUC values were greater than 0.6, as detailed in S5A–S5J Fig.

The gene biomarkers were tested using the Wilcoxon rank-sum test on the GSE113439 dataset, as shown in Fig 9. Seven genes (PTGS2, TBXA2R, GPR171, KLC3, MSRA, LYSMD3, and CMTM5) showed significant differences between the disease and control groups.

**Table 3. The gene biomarkers by the proposed method.**

| Gene ID | Symbol | Full Name |
|---------|--------|-----------|
| 5743 | PTGS2 | Prostaglandin-endoperoxide synthase 2 |
| 27334 | P2RY10 | P2Y receptor family member 10 |
| 6915 | TBXA2R | Thromboxane A2 receptor |
| 94039 | ZNF101 | Zinc finger protein 101 |
| 29909 | GPR171 | G protein-coupled receptor 171 |
| 147700 | KLC3 | Kinesin light chain 3 |
| 3759 | KCNJ2 | Potassium inwardly rectifying channel subfamily J member 2 |
| 4482 | MSRA | Methionine sulfoxide reductase A |
| 116068 | LYSMD3 | LysM domain containing 3 |
| 116173 | CMTM5 | CKLF like MARVEL transmembrane domain containing 5 |

**Table 4. Significance levels of gene biomarkers.**

| Gene Symbol | Significant level | P_value |
| --- | --- | --- |
| PTGS2 | *** | 2.29E-05 |
| P2RY10 | *** | 3.44E-05 |
| TBXA2R | *** | 7.42E-09 |
| ZNF101 | *** | 6.33E-10 |
| GPR171 | *** | 7.32E-05 |
| KLC3 | *** | 2.10E-10 |
| KCNJ2 | ** | 1.28E-03 |
| MSRA | *** | 1.09E-09 |
| LYSMD3 | *** | 2.85E-05 |
| CMTM5 | *** | 7.29E-11 |

*indicates significance at the 0.05 level, **indicates significance at the 0.01 level, and ***indicates significance at the 0.001 level.

## 4. Discussion

Identifying biomarkers based on gene expression data has been a hot topic. We proposed a new algorithm, DeepGCFS, for identifying hypertension biomarkers. This algorithm constructs a graph structure incorporating real gene dependency relationships, builds a GNN model, and designs a loss function based on link prediction and self-supervised learning ideas for training, obtaining feature vectors with global information representation capability. Then, the gene modules were partitioned using the HCM algorithm, with SI, CHI, and DBI serving as evaluation metrics. The clustering performance of the proposed method was found to surpass that of the other methods. Finally, ten biomarkers were identified through an integrated feature selection approach.

We found that six out of the ten identified biomarkers are reported to be associated with hypertension disease. Although there is currently no research reporting the association of the genes P2RY10, GPR171, KLC3, and LYSMD3 with hypertension, these genes may indirectly affect the occurrence of hypertension. PTGS2, also known as Cox-2, is a critical enzyme in prostaglandin biosynthesis, with dual functions as a cyclooxygenase and a peroxidase. It is

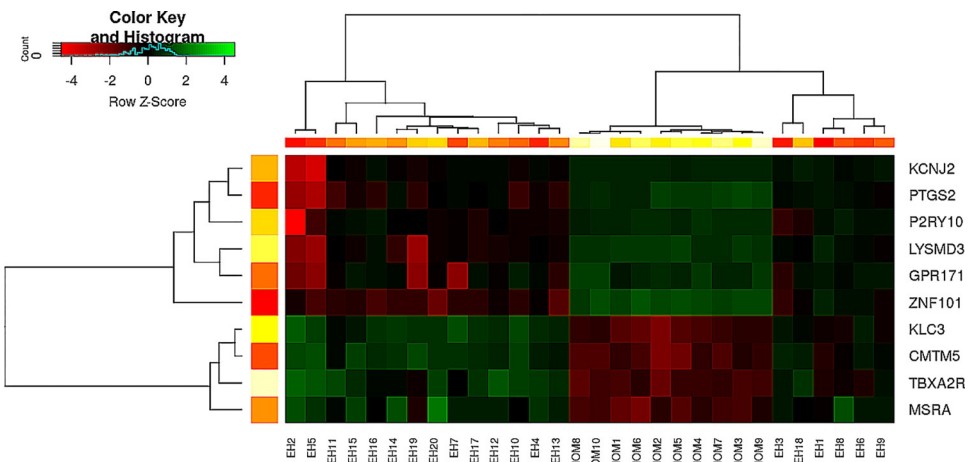

**Fig 5. Heat map analysis of the biomarkers.**

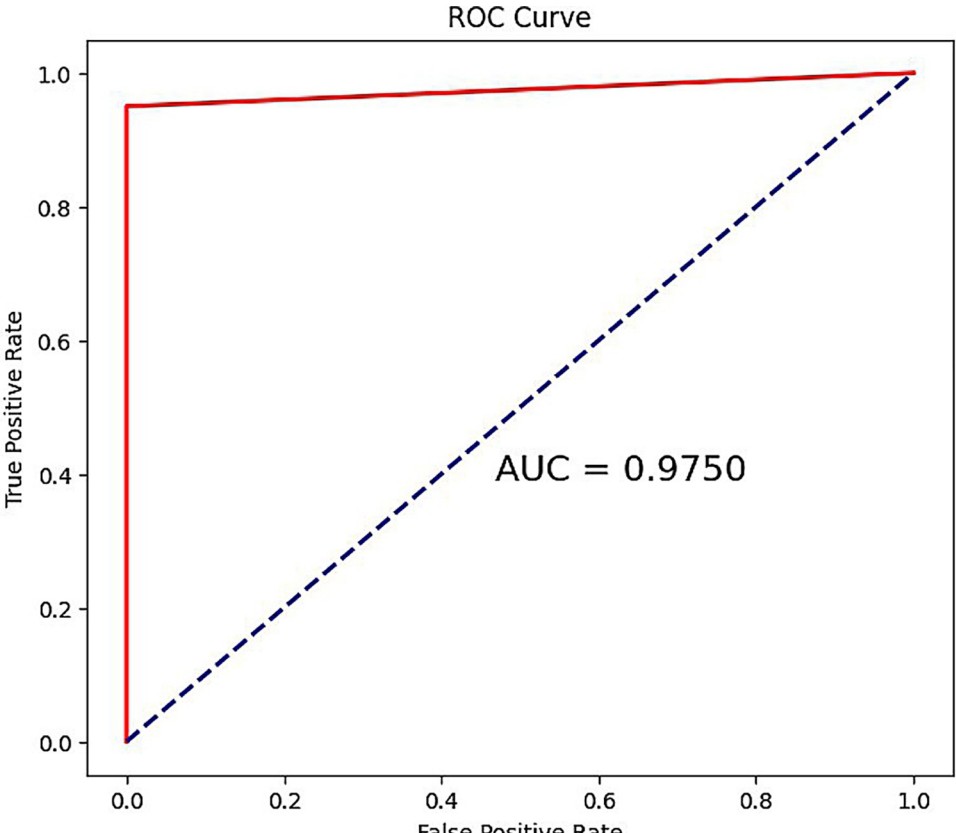

**Fig 6. ROC for all gene biomarkers as features.**

pivotal in prostaglandin biosynthesis associated with inflammation, cell proliferation, and angiogenesis. Current research has shown a correlation between PTGS2 and hypertension. For example, a study has indicated that inhibiting mTORC1 activity in endothelial cells can reduce the production of prostaglandin E2 and lead to hypertension by lowering YAP-mediated COX-2/mPGES-1 expression[31]. Elizabeth et al.[32] conducted a meta-analysis and found that COX-2 inhibitors may increase the risk of cardiovascular adverse events, such as hypertension, heart failure, and edema. Additionally, numerous drugs target PTGS2 for intervention in the treatment of hypertension and other cardiovascular diseases. For example, some nonsteroidal anti-inflammatory drugs (NSAIDs) can inhibit PTGS2, reducing inflammatory responses and lowering blood pressure[33]. TBXA2R (Thromboxane A2 Receptor) is a member of the thromboxane A2 receptor family that encodes G protein-coupled receptors. This protein interacts with thromboxane A2 to induce platelet aggregation and regulate hemostasis.

**Table 5. Comparison with published methods.**

| Literature | Method | AUC |
|---|---|---|
| Gao[3] | DEG+cytoHCA | 90.75% |
| Li[5] | WGCNA | 92.50% |
| Jiang[9] | DEG+SVM+Lass0 | 95.00% |
| Errington[30] | HybridFS | 87.50% |
| DeepGCFS | DeepGCFS | 97.50% |

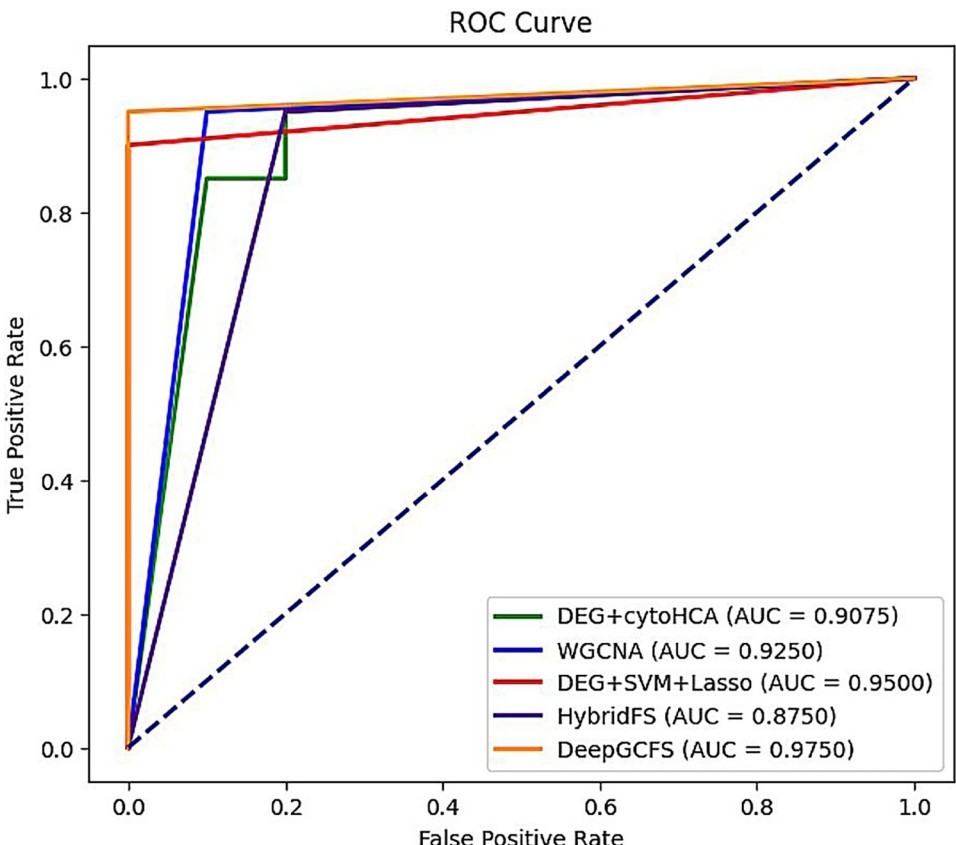

**Fig 7. The ROC Curve of Comparison with published methods.**

TXA2 binding to the glomerular TP receptor in the kidneys causes intense vasoconstriction. It has been reported that TBXA2R rs13306046 is associated with blood pressure[34]. Studies have shown that hyperglycemia activates thromboxane A2 receptors to increase DOCA-induced hypertension in rats through PTEN-Ak[35]. ZNF101 is a protein-coding gene that binds to nucleic acids and performs numerous crucial functions. GO annotations suggest that the protein encoded by this gene possesses nucleic acid binding functionality. Research has shown a significant association between the ZNF101 rs2304130 and hypertension[36]. The protein encoded by the KCNJ is a transmembrane protein and an inward rectifying potassium channel. Intracellular ATP activates it and may play a significant role in potassium homeostasis. Several studies have indicated an association between this gene and some members of its gene family with hypertension. Research suggests that the KCNJ1 may play a role in sodium handling or blood pressure regulation in the renal medulla[37]. A study has found that the KCNJ5 rs2604204 is associated with elevated plasma aldosterone levels in newly diagnosed, untreated hypertensive patients and is also linked to increased levels of plasma renin, angiotensin I, and angiotensin II[38]. Knockout of the KCNJ11 may lead to maladaptive remodeling and heart failure in hypertensive patients[39]. Wang et al.[40] identified KCNJ11 E23K and KCNMB1 E65K as potential susceptibility factors for primary hypertension through mate-analysis. The KCNJ11 rs5219 may be associated with susceptibility to primary hypertension in the Kazakh population in Xinjiang[41]. MSRA encodes a ubiquitous and highly conserved protein that reduces methionine sulfoxylase to methionine. The function of this protein is to repair proteins damaged by oxidation and has an important function for proteins inactivated

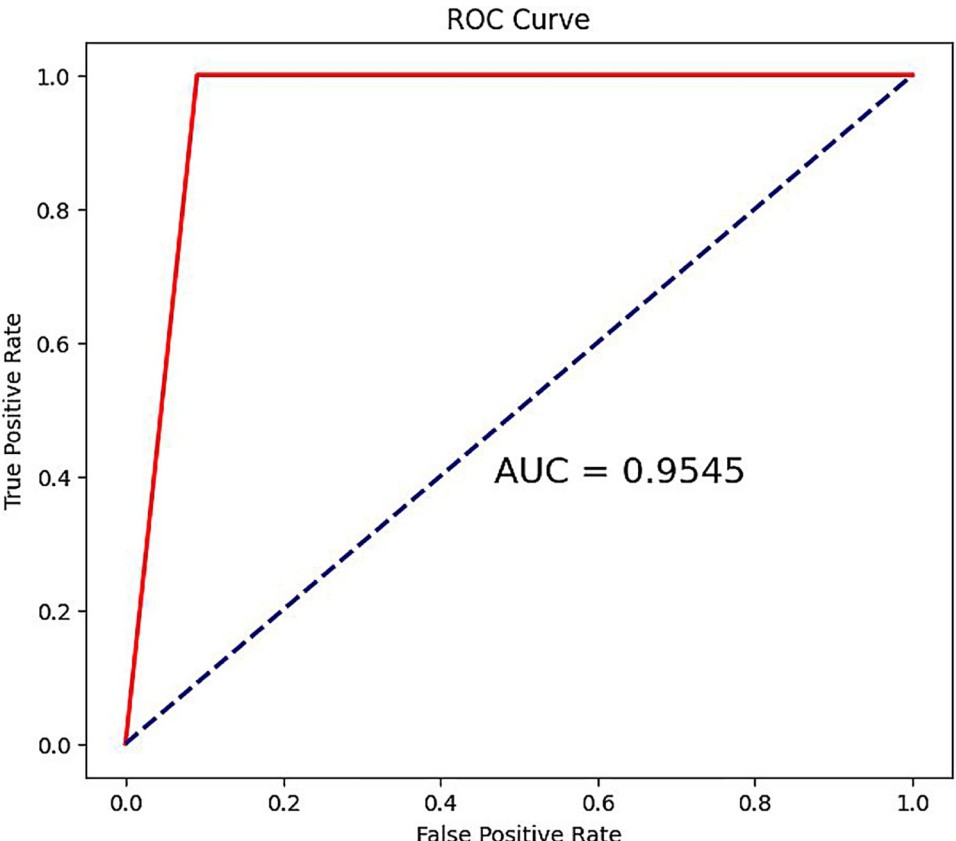

**Fig 8. ROC for all gene biomarkers as features on the GSE113439 set.**

by oxidation. Studies have reported that MSRA and its polymorphisms are related to dyslipidemia, susceptibility to cardiovascular diseases such as atherosclerosis, vascular smooth muscle cell and intimal hyperplasia, and blood pressure regulation[42–45]. CMTM5 encodes a member of the chemokine superfamily. This gene family encodes multichannel membrane proteins, similar to the transmembrane four superfamily of chemokines and signaling molecules. The encoded protein may exhibit tumor suppressor activity. Research has shown that CMTM5

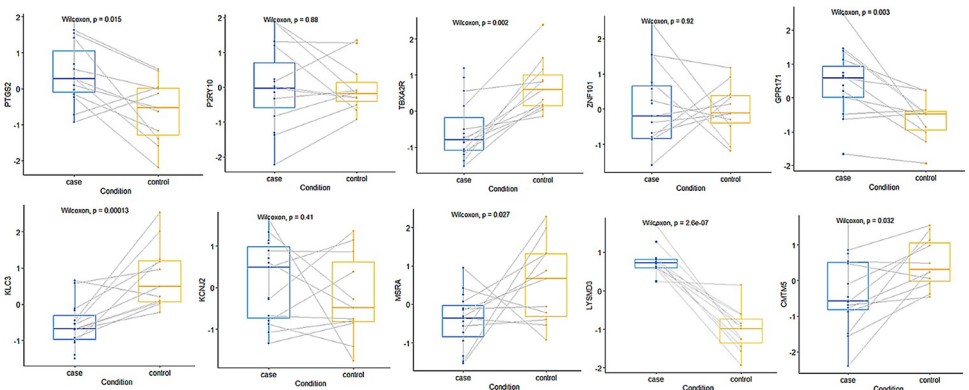

**Fig 9. The distribution of the gene biomarkers in positive and negative samples in the GSE113439 database.**

exerts anti-atherosclerotic effects by inhibiting migration and proliferation of vascular smooth muscle cells[46].

All ten biomarkers were significantly differentiated by t-test and heat map analysis. Using AUC as an evaluation metric, a high classification performance of 97.50% was achieved. Each gene has features, and all can exhibit strong classification capability for the samples. Using the GSE113439 dataset for external validation, ten gene biomarkers were used as features, and they classified samples well with an AUC value of 95.45%. Further, the Wilcoxon rank-sum test on the ten gene biomarkers revealed significant differences in seven genes (LYSMD3, TBXA2R, KLC3, GPR171, PTGS2, MSRA, and CMTM5). These can demonstrate the superiority of the proposed algorithm, effectively selecting potential hypertension biomarkers.

## 5. Conclusion

This paper proposes a novel hypertension biomarker identification algorithm, DeepGCFS. The algorithm effectively uses the dependency between genes and uses GNN for gene representation learning. Module detection via hybrid clustering and integrated feature selection methods were used to identify hypertension biomarkers. The results demonstrate that the biomarkers identified by DeepGCFS exhibit significant biological. Additionally, the classification performance was evaluated, further proving the effectiveness and reliability of the selected features. However, a limitation of this study is the lack of biological validation for the identified biomarkers. In future work, we plan to continue our research by incorporating experimental validation to further confirm the biological relevance of our findings and enhance their practical applicability.

## Supporting information

**S1 Fig. GO analysis bubble plot for other four modules.**
(TIF)

**S2 Fig. KEGG analysis bubble plot for other four module.**
(TIF)

**S3 Fig. Visualization of interaction between genes in other four module.**
(TIF)

**S4 Fig. ROC curve for gene biomarkers.**
(TIF)

**S5 Fig. ROC curve for gene biomarkers on the validated set.**
(TIF)

## Acknowledgments

We acknowledge SRA and GEO databases for providing their platforms and contributors for uploading meaningful datasets.

## Author Contributions

**Conceptualization:** Zongjin Li, Xiaoming Wu, Changxin Song.

**Formal analysis:** Zongjin Li, Liqin Tian, Libing Bai.

**Investigation:** Zongjin Li, Zeyu Jia.

**Methodology:** Zongjin Li, Liqin Tian, Xiaoming Wu.

**Software:** Zongjin Li, Liqin Tian, Libing Bai, Zeyu Jia.

**Supervision:** Zongjin Li.

**Validation:** Zongjin Li, Libing Bai.

**Visualization:** Zongjin Li, Changxin Song.

**Writing – original draft:** Zongjin Li, Xiaoming Wu.

**Writing – review & editing:** Zongjin Li, Changxin Song.

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
