## [Decision Letter · Decision Letter 0]

23 Jan 2024

PONE-D-23-39381Identification of Hypertension Gene Expression Biomarkers Based on the DeepGCFS AlgorithmPLOS ONE

Dear Dr. Li,

Thank you for submitting your manuscript to PLOS ONE. After careful consideration, we feel that it has merit but does not fully meet PLOS ONE’s publication criteria as it currently stands. Therefore, we invite you to submit a revised version of the manuscript that addresses the points raised during the review process.

We look forward to receiving your revised manuscript.

Kind regards,

Babatunde Olanrewaju Motayo, Ph.D.

Academic Editor

PLOS ONE

“This research was supported by the Hebei Internet of Things Monitoring Technology Innovation Center(21567693H), the Qinhai IoT Key Laboratory(No.2017-ZJ-Y21), the Fundamental Research Funds for the Central Universities(3142021009), China University industry research innovation Fund (2021DA12008), China National Key Research and Development (No. 2018YFC0808306), and Tibetan Information Processing and Machine Translation Key Laboratory of Qinghai Province (2020-ZJ-Y05).”

“This research was supported by the Hebei Internet of Things Monitoring Technology Innovation Cen-ter(21567693H), the Qinhai IoT Key Laboratory(No.2017-ZJ-Y21), the Fundamental Research Funds for the Central Universities(3142021009), China University industry research innovation Fund (2021DA12008), China National Key Research and Development (No. 2018YFC0808306), and Ti-betan Information Processing and Machine Translation Key Laboratory of Qinghai Province (2020-ZJ-Y05).”

“This research was supported by the Hebei Internet of Things Monitoring Technology Innovation Center(21567693H), the Qinhai IoT Key Laboratory(No.2017-ZJ-Y21), the Fundamental Research Funds for the Central Universities(3142021009), China University industry research innovation Fund (2021DA12008), China National Key Research and Development (No. 2018YFC0808306), and Tibetan Information Processing and Machine Translation Key Laboratory of Qinghai Province (2020-ZJ-Y05).”

Reviewers' comments:

Reviewer's Responses to Questions

**Comments to the Author**

1. Is the manuscript technically sound, and do the data support the conclusions?

Reviewer #1: Partly

Reviewer #2: Partly

2. Has the statistical analysis been performed appropriately and rigorously? 

Reviewer #1: Yes

Reviewer #2: No

3. Have the authors made all data underlying the findings in their manuscript fully available?

Reviewer #1: No

Reviewer #2: No

4. Is the manuscript presented in an intelligible fashion and written in standard English?

Reviewer #1: Yes

Reviewer #2: No

5. Review Comments to the Author

Reviewer #1: In this study, the authors introduce the Deep Graph Clustering Feature Selection (DeepGCFS) algorithm to enhance the identification of hypertension biomarkers. The algorithm employs a graph network to portray gene relationships, integrates link prediction and self-supervised learning, and incorporates a hybrid clustering method with integrated feature selection. However, some drawbacks exist:

1. In the introduction, discuss the advantages and disadvantages of the reviewed literature methods for identifying hypertension biomarkers.

2. The phrase "They also do not take into account the attribute information of features" means that existing methods may overlook the specific characteristics or attributes of features when identifying biomarkers. This needs more clarification on how these attributes are important.

3. Expand on the sentence about feature selection methods, detailing various types such as wrapper and filter selections. Highlight that while wrapper methods capture some feature relationships through model performance, they may not explicitly model dependencies among features, which is crucial in biomarker identification.

4. Strengthen the literature review by including key publications on predicting marker genes associated with hypertension, such as the works by Gao et al. and Jiang et al., emphasizing their methodologies and findings.

1. Gao Y, Qi GX, Jia ZM, Sun YX. Prediction of marker genes associated with hypertension by bioinformatics analyses. Int J Mol Med. 2017 Jul;40(1):137-145. doi: 10.3892/ijmm.2017.3000. Epub 2017 May 25. PMID: 28560446; PMCID: PMC5466388.

2. Jiang C, Jiang W. Lasso algorithm and support vector machine strategy to screen pulmonary arterial hypertension gene diagnostic markers. Scott Med J. 2023 Feb;68(1):21-31. doi: 10.1177/00369330221132158. Epub 2022 Oct 17. PMID: 36253715.

5. In the introduction, elaborate on the proposed approach's foundation in deep graph clustering. Discuss various clustering methods and their pros and cons, including Deepwalk, Node2vec, SDNE, and Weighted Correlation Network Analysis (WGCNA).

6. Explain the criteria used for selecting the highest correlations among all pairs of genes. Specify the threshold considered as "highest" and clarify the process of creating the graph network by merging relevant edges (eGp and eGs).

7. Enhance clarity in the methodology by providing a detailed pseudocode, step by step, color each stage. Encourage sharing the code and dataset on GitHub for transparency.

8. Explain why Mutual Information Feature Selection (MIFS) is used in the final step, providing justification for choosing it over other feature ranking methods.

9. Compare the proposed approach's performance with at least two gene prioritization methods, considering AUC and Gene Ontology (GO) analysis.

10. In GO and KEGG pathway analyses, please perform separate assessments for up-regulated and down-regulated genes.

11. Utilize commonly used cross-validation (5 or 10 folds) instead of 3. Draw a heatmap of identified biomarkers and provide details on the number of genes and edges in the first step.

12. Compare the performance of the proposed approach with WGCNA.

13. For visualizing gene modules, please utilize the widely-used STRING platform, incorporating color-coding to distinguish between up-regulated and down-regulated genes.

Reviewer #2: Dataset:

The dataset used for this problem - a graph-theoretic machine learning analysis to identify biomarkers of hypertension - seems to be rather small - just a handful of HT and normal samples. A larger dataset may be necessary; please see comment on benchmarking.

Validation:

The authors have used a GEO dataset for validation, but have performed three-fold cross-validation with the biomarkers. This is not valid. Even the biomarkers do not replicate on the validation dataset.

There is no train-test split either, and the authors may consult with a statistician for proper validation of their models.

Benchmarking with published models:

Many models exist for prediction of hypertension. The authors have not compared their results with any of the existing models. Existing methods might perform better, this may be studied.

Reproducibility:

No stepwise results have been shared, and data supporting the conclusions has not been shared. No models have been shared, and work cannot be reproduced at all. Even the authors' biomarker replication efforts do not succeed with the validation dataset.

6. PLOS authors have the option to publish the peer review history of their article (what does this mean?). If published, this will include your full peer review and any attached files.

Reviewer #1: No

Reviewer #2: **Yes: **Ashok Palaniappan

---

## [Author Response · Author response to Decision Letter 0]

14 Apr 2024

Response Letter

(Manuscript ID: PONE-D-23-39381)

Dear Babatunde OlanrewajuMotayo, Ph.D. and Reviewers,

Thanks very much for taking your time to review our manuscript (Titile: Identification of Hypertension Gene Expression Biomarkers Based on the DeepGCFS Algorithm). We really appreciate all your comments and suggestions! In this revision, we have addressed all of these comments. We hope the revised manuscript has now met the publication standard of PLOS ONE. 

Our point-to-point responses to the comments raised by the reviewers and the additional editor are listed in reponse letter(Response to Reviewers.pdf).

Thanks again!

Yours Sincerely,

Zongjin Li

E-mail: 202033341012@stu.qhnu.edu.cn

---

## [Decision Letter · Decision Letter 1]

9 Sep 2024

PONE-D-23-39381R1Identification of Hypertension Gene Expression Biomarkers Based on the DeepGCFS AlgorithmPLOS ONE

Dear Dr. Li,

Thank you for submitting your manuscript to PLOS ONE. After careful consideration, we feel that it has merit but does not fully meet PLOS ONE’s publication criteria as it currently stands. Therefore, we invite you to submit a revised version of the manuscript that addresses the points raised during the review process.

**You could find the comments by the reviewers, and the decision to this paper is Major Revise. We advise the authors revised this paper as the reviewers suggested, and give a point-to-point response for the revised version. We are glad to receive your new version of this work.**

We look forward to receiving your revised manuscript.

Kind regards,

Hui Li

Academic Editor

PLOS ONE

Reviewers' comments:

Reviewer's Responses to Questions

**Comments to the Author**

1. If the authors have adequately addressed your comments raised in a previous round of review and you feel that this manuscript is now acceptable for publication, you may indicate that here to bypass the “Comments to the Author” section, enter your conflict of interest statement in the “Confidential to Editor” section, and submit your "Accept" recommendation.

Reviewer #2: (No Response)

Reviewer #3: All comments have been addressed

2. Is the manuscript technically sound, and do the data support the conclusions?

Reviewer #2: Partly

Reviewer #3: Yes

3. Has the statistical analysis been performed appropriately and rigorously? 

Reviewer #2: No

Reviewer #3: (No Response)

4. Have the authors made all data underlying the findings in their manuscript fully available?

Reviewer #2: No

Reviewer #3: No

5. Is the manuscript presented in an intelligible fashion and written in standard English?

Reviewer #2: No

Reviewer #3: Yes

6. Review Comments to the Author

Reviewer #2: With respect to validation, the authors mention that "Response #2：Thank you for the reviewers' suggestions. Based on the sample size and recommendations from statisticians, we adopted the Leave-one-out cross-validation ".

LOOCV remains a form of cross-validation, which is unsuited for external validation. The model already built need not be cross-validated. How does that make sense?

With respect to benchmarking, the authors mention that "Response #3 ： Thank you for the reviewers' suggestions. We have conducted a performance comparison analysis of our proposed method with other methods in the literature. The experimental results are presented in the table."

The authors may be referring to Table 5. From Table 5, it is clear that many previous methods had been developed for this problem. Benchmarking all these models requires a common dataset for reference and evaluation. It is is not clear from the presentation if all the models were evaluated on the same dataset. This may be asserted if true. If not, then it is not meaningful to present this comparison.

Thanks and good wishes.

Reviewer #3: In the revised version, my former comments have been addressed. I have no critical comments except that the wet-lab experiments for validating these findings. I think the authors will conduct them in the near future and they discuss this important issue in this paper.

7. PLOS authors have the option to publish the peer review history of their article (what does this mean?). If published, this will include your full peer review and any attached files.

Reviewer #2: **Yes: **Ashok Palaniappan

Reviewer #3: No

---

## [Author Response · Author response to Decision Letter 1]

3 Oct 2024

Response Letter

(Manuscript ID: PONE-D-23-39381R1)

Dear Hui Li and Reviewers,

Thank you very much for taking the time to review our manuscript (Title: Identification of Hypertension Gene Expression Biomarkers Based on the DeepGCFS Algorithm [PONE-D-23-39381R1]). We sincerely appreciate all your insightful comments and suggestions. In this revision, we have addressed all the points raised. We hope that the revised manuscript now meets the publication standards of PLOS ONE. The main corrections in the paper and our responses to the reviewer’s comments are as follows:

REVIEWER #2

Comment #1:LOOCV remains a form of cross-validation, which is unsuited for external validation. The model already built need not be cross-validated. How does that make sense?

Response #1: Thank you very much for your valuable comments and for taking the time to review our revised manuscript. We understand your concerns regarding the use of LOOCV and its limitations for external validation. To clarify, our external dataset was not used to validate the model itself but rather to assess the identified gene biomarkers. Specifically, our aim was to confirm that the biomarkers identified in our study are also hub genes in other independent datasets. Due to the relatively small size of the external dataset, we employed the leave-one-out cross-validation (LOOCV) method to evaluate the classification performance of the identified biomarkers. LOOCV enabled us to fully utilize the available data and ensure the reliability of the classification results, even with limited samples. That is, this process was focused on validating the consistency of the identified biomarkers, not the model. We hope this clarifies our approach and addresses your concerns. Thank you again for your thoughtful feedback.

Comment #2:The authors may be referring to Table 5. From Table 5, it is clear that many previous methods had been developed for this problem. Benchmarking all these models requires a common dataset for reference and evaluation. It is is not clear from the presentation if all the models were evaluated on the same dataset. This may be asserted if true. If not, then it is not meaningful to present this comparison.

Response #2: Thank you for your valuable comment. We agree that it is essential for a fair comparison to benchmark all models on the same dataset. In response to your concern, we would like to clarify that all the models presented in Table 5 were indeed evaluated on the same dataset. To improve the clarity of this point, we have revised the manuscript to explicitly state this fact in the relevant section. The following sentence has been added to the revised manuscript: “All the models listed in Table 5 have been benchmarked on the same dataset, ensuring consistency and fairness in the comparison” (in 3.8 Algorithm Performance Comparison). We hope this revision addresses your concern and clarifies the methodology used for the model evaluation in our study.

Special thanks to you for your good comments！

REVIEWER #3

Comment: In the revised version, my former comments have been addressed. I have no critical comments except that the wet-lab experiments for validating these findings. I think the authors will conduct them in the near future and they discuss this important issue in this paper.

Response: 

Thank you very much for taking the time to review our revised manuscript and for your positive feedback. We are pleased to hear that our revisions have addressed your concerns. We fully agree with your assessment regarding the importance of wet-lab experiments to validate our findings. As you correctly pointed out, these experiments are essential for confirming the biological relevance of our results. In future work, we plan to continue our research by incorporating experimental validation to further confirm the biological relevance of our findings. We sincerely appreciate your understanding and constructive feedback.

Special thanks to you for your good comments！

Thanks again!

Yours Sincerely,

Zongjin Li

E-mail: 202033341012@stu.qhnu.edu.cn

---

## [Decision Letter · Decision Letter 2]

8 Nov 2024

Identification of Hypertension Gene Expression Biomarkers Based on the DeepGCFS Algorithm

PONE-D-23-39381R2

Dear Dr. Li,

We’re pleased to inform you that your manuscript has been judged scientifically suitable for publication and will be formally accepted for publication once it meets all outstanding technical requirements.

Kind regards,

Hui Li

Academic Editor

PLOS ONE

Additional Editor Comments (optional):

Reviewers' comments:

Reviewer's Responses to Questions

**Comments to the Author**

1. If the authors have adequately addressed your comments raised in a previous round of review and you feel that this manuscript is now acceptable for publication, you may indicate that here to bypass the “Comments to the Author” section, enter your conflict of interest statement in the “Confidential to Editor” section, and submit your "Accept" recommendation.

Reviewer #4: (No Response)

2. Is the manuscript technically sound, and do the data support the conclusions?

Reviewer #4: (No Response)

3. Has the statistical analysis been performed appropriately and rigorously? 

Reviewer #4: (No Response)

4. Have the authors made all data underlying the findings in their manuscript fully available?

Reviewer #4: (No Response)

5. Is the manuscript presented in an intelligible fashion and written in standard English?

Reviewer #4: (No Response)

6. Review Comments to the Author

Reviewer #4: The paper is well-written, the novelty is sufficient, and the response to the reviewer was satisfactory.

7. PLOS authors have the option to publish the peer review history of their article (what does this mean?). If published, this will include your full peer review and any attached files.

Reviewer #4: **Yes: **Pouya Bolourchi

---

## [Editor Report · Acceptance letter]

22 Dec 2024

PONE-D-23-39381R2 

PLOS ONE

Dear Dr. Li, 

I'm pleased to inform you that your manuscript has been deemed suitable for publication in PLOS ONE. Congratulations! Your manuscript is now being handed over to our production team.

Kind regards, 

on behalf of

Professor Hui Li 

Academic Editor

PLOS ONE